# Is the fire even bigger? Burnout in 800 medical and nursing students in a low middle income country

Syed Hamza Mufarrih[1,2]*, Nada Qaisar Qureshi[1,2], Syeda Amrah Hashmi[2], Abbas Raza Syed[2], Muhammad Zohaib Anwar[2], Riaz Hussain Lakdawala[3], Nargis Asad[4], Adil Haider[5], Shahryar Noordin[3]

1 Department of Medicine, University of Kentucky, Bowling Green, Kentucky, United States of America, 2 Medical College, Aga Khan University Medical College, Karachi, Pakistan, 3 Department of Surgery, Aga Khan University Hospital, Karachi, Pakistan, 4 Department of Psychiatry, Aga Khan University Hospital, Karachi, Pakistan, 5 Dean, Aga Khan University Medical College, Karachi, Pakistan

* Shmu225@uky.edu, hamzamufarrih@live.com

**Data Availability Statement:** The dataset is restricted in accordance with the guidelines set by the Aga Khan University Ethics Committee due to

## Abstract

### Background

Burnout, characterized by emotional exhaustion (EX), depersonalization (DP), and a reduced sense of personal efficacy (PF) among medical and nursing students can lead to suicidal ideation, lack of empathy, and dropouts. Previous studies have used over-simplified definitions of burnout that fail to capture its complexity. We describe the prevalence of burnout profiles and its risk factors among medical and nursing students.

### Methods

A cross sectional study was conducted at a tertiary care University Hospital in Pakistan. The Maslach Burnout Inventory (MBI) survey was disseminated via SurveyMonkey over a period of 4 months (November 2019 to February 2020) to 482 Medical and 441 nursing students. The MBI tool measures the dimensions of EX, DP, and PF to describe seven burnout profiles. Multivariable regression was used to identify predictors of burnout.

### Results

The response rate was 92% in nursing and 87.3% in medical students. The prevalence of burnout in medical and nursing students was 16.9% and 6.7% respectively (p<0.001), with 55.7% (n = 427) suffering from at least one burnout profile. Only 32.5% (n = 250) students felt engaged, (42.3% medical, 22.7% nursing students, p<0.001). The most common profile was ineffective (32.5%, n = 250), characterized by a reduced sense of personal efficacy (35.6% medical, 29.4% nursing students; p = 0.065). Medical students were at higher risk of burnout compared to nursing students (OR = 2.49 [1.42, 4.38]; p<0.001) with highest risk observed in year 4 (OR = 2.47 [1.02, 5.99]; p = 0.046). Other risk factors for burnout included occasional drug use (OR = 1.83 [1.21, 8.49]; p = 0.017) and living in a hostel (OR = 1.64 [1.01,2.67]; p = 0.233).

the sensitive nature of its content, which holds the potential for participant identification despite de-identification efforts. Researchers who meet the confidentiality criteria may request access to the data by contacting the Department of Surgery's Data Access Committee at Aziz Hayder. [aziz.hayder@aku.edu].

**Funding:** The author(s) received no specific funding for this work.

**Competing interests:** The authors have declared that no competing interests exist.

## Conclusion and relevance

Two-thirds of our participants experienced at least one dimension of burnout with the highest prevalence of a reduced sense of PF. Drivers of burnout unique to a lower-middle-income country need to be understood for effective interventions. Faculty training on principles of student evaluation and feedback may be beneficial.

## Introduction

Burnout, characterized by emotional, physical, and mental exhaustion, is a pervasive condition with far-reaching implications. Maslach *et al.* delineated the three burnout dimensions: emotional exhaustion (EX), depersonalization (DP) and a reduced sense of personal efficacy (PF) [1]. It is a construct distinct from other mental disorders or general stress reactions, manifesting with many physical and psychological consequences including depression, anxiety, insomnia, and metabolic diseases such as diabetes and metabolic syndrome [2, 3].

Medical health professionals, grappling with extended work hours, long years of training, limited life outside of work, and societal expectations, are at a higher risk of burnout [2, 4–7]. The intricate dynamics of student life may exacerbate the above circumstances. The pressures of competition, uncertainty regarding admissions into further educational programs, social navigation and concern for reputations and rapport with authority figures contribute to a student's vulnerability to burnout [8–12].

Students with burnout have frequently been reported to have thoughts of dropping out [13], diminished empathy [14] and even suicidal ideation [15]. Therefore, it is crucial to identify and address the ongoing stressors associated with burnout to mitigate its adverse consequences.

Previous studies from the United States [11], Sweden [9], Korea [16], Brazil [17], China [18], Hong Kong [19] and UK [20] have reported disparate prevalance rates and gender disparities in burnout amongst medical and nursing students, suggesting the influence of geographical and/or cultural factors. The majority of these studies have inappropriately applied the Maslach Burnout Inventory (MBI), using unified cut-off scores or single-item. These oversimplify the concept of burnout, thereby limiting the understanding of this complex syndrome. Using the results of the MBI-burnout tool as a continuous variable oversimplifies the burnout phenomenon, potentially overlooking critical nuances between the dimensions of exhaustion, cynicism, and professional efficacy that distinctly impact individuals' experiences and responses to interventions. Previous studies on student burnout from lower-middle-income countries have used unified cut-off scores without expanding on various burnout dimensions. [21]

Our study aimed to assess and compare the prevalence of burnout, and its various dimensions (EX, DP, and PF) as described by the MBI guidelines, among medical and nursing students at a tertiary care university hospital in Pakistan.

## Methods

### Study design & participants

A cross-sectional study was conducted at a tertiary care university hospital in Pakistan over 4 months from November 2019 to February 2020. All students enrolled at the medical (n = 482)

and nursing (n = 441) school were eligible to participate. The methodology of this study has been discussed in detail in a previously published protocol [22].

## Eligibility criteria

The study population included medical and nursing students enrolled at Aga Khan University Hospital, Karachi, Pakistan, from November 2019 to February 2020. Those who did not consent, and participants with one or more incomplete items on the 16-item burnout surveys were excluded. Incomplete demographic information did not warrant exclusion.

## Data collection

A survey comprising a consent form, demographic questions, and the 16-item *MBI—General* Survey for Students was emailed to eligible students via SurveyMonkey. Students were informed that participation was voluntary and anonymous. Informed consent was obtained through an online form at the survey's outset. Students who did not consent were removed from the mailing list. Non-respondents received bi-monthly reminder emails.

The survey was incentivized with meal coupons, provided by the medical school's dean's office, awarded to every fifth participant. Additionally, students had the option to opt-in for individual score reports.

## MBI—General Survey for Student tool

*The MBI–General Survey for Students* is a 16-item questionnaire designed to assess university students' attitudes toward their studies and their reactions to academic work. The MBI tool is considered a criterion standard [21, 23]. The responses are evaluated across three dimensions of burnout: Emotional Exhaustion (EE), Depersonalization (DP), and Personal Efficacy (PF). Emotional Exhaustion describes feelings of being emotionally overextended by one's academic work. Depersonalization assesses a detached response toward one's studies, and personal accomplishment evaluates feelings of competence and successful achievement in one's work.

Using responses from the 16-point questionnaire, the individual scores for the EX, DP, and PF subscales were calculated as continuous variables. The MBI-provided guidelines were then applied to the three dimensions of burnout to determine cut-off values for categorizing levels of each dimension as high or low (details discussed subsequently) [21].

- Cut-off for emotional exhaustion = z ee = Mean + SD (0.5)

- Cut-off for cynicism/ depersonalization = z cy = Mean + SD (1.25)

- Cut-off for Personal accomplishment = z pa = Mean + SD (0.10)

Based on these cut-offs, individuals were categorized as "high" or "low" for each subscale. Combinations of "high" and "low" scores in the three burnout dimensions are used to describe the seven burnout profiles (burnout, engaged, overextended, ineffective, disengaged, disengaged and ineffective, overextended and ineffective) as outlines in S1 and S2 Tables [21]. Individuals are classified as experiencing high levels of burnout if they score *high* on EE, *high* on DP, and either *high* or *low* on PA, per guidelines outlined by MBI [1, 21] (S1 Table).

The reliability of this tool in South Asian population is supported by Cronbach coefficient of 0.90 for EX, 0.79 for DP and 0.71 for PF [26] Permission for reuse of the *(MBI)–General Survey for* Students for obtained from Mind Garden, Inc. on July 11, 2019.

## Sample size calculation

Given that the survey targets every medical and nursing student, this cross-sectional study employed a census approach, eliminating the need for traditional sample size calculations.

## Study outcome measures

The primary objective of this study was to determine the prevalence of burnout in medical and nursing students at Aga Khan University, Karachi Pakistan. The secondary objective was to identify risk factors associated with burnout in each group together and separately.

## Statistical analysis

IBM Corp. SPSS Statistics (version 29.0) was used to perform the study analyses. The Shapiro-Wilk test was used to assess the normality of all baseline demographic variables in relation to burnout. Since our data followed a normal distribution, central tendencies were reported as mean ± standard deviation, and a t-test was utilized to compare the variables. For binary variables, the chi-square test was used for comparison. All statistical tests were two-sided, with a significance level (type 1 error rate) set at 0.05. The reliability of each dimension (EX, DP, and PF) for use in our study population was calculated using Cronbach's alpha.

Bivariate logistic regression was performed to study the relationship between burnout and demographic variables. For univariate analysis, we used a significance level of $p<0.25$, while for multivariable analysis, a significance level of 0.05 was used as the entry threshold. Following univariate analysis, multivariable regression was performed for variables meeting the criteria to identify independent predictors of burnout among the study participants.

**Ethical approval.** The study was approved by the Institutional review board (IRB) at the Aga Khan University Hospital, Karachi, Pakistan.

# Results

## Characteristics of the responders (Table 1)

The survey was disseminated to 482 medical students and 441 nursing students. The mean age of our study population was 21.30 ± 1.84, with 70.5% female. The majority (98.6%) were single

**Table 1. Demographic and characteristics of study population.**

| | Overall (n = 767) | Medical student (*n = 379*) | Nursing Student (*n = 388*) | *P-value* |
|---|---|---|---|---|
| **Age (mean ± SD)** | 21.30 ± 1.84 | 21.59 ± 1.70 | 21.01 ± 1.94 | **< 0.001** |
| **Female n (%)** | 541 (70.5) | 193 (50.9) | 348 (89.7) | **< 0.001** |
| **Marital Status n (%)** | | | | 0.105 |
| Single | 756 (98.6) | 377 (99.5) | 379 (97.7) | |
| Married | 10 (1.3) | 2 (0.5) | 8 (2.1) | |
| Widowed | 1 (0.1) | 0 (0) | 1 (0.3) | |
| **Have children n (%)** | 5 (0.7) | 0 (0) | 5 (1.3) | **0.027** |
| **Hostel residence n (%)** | 467 (60.9) | 200 (52.8) | 267 (68.8) | **<0.001** |
| **Cigarette smoking n (%)** | 44 (5.7) | 29 (7.7) | 15 (3.9) | **0.037** |
| **Substance Use n (%)** | | | | **0.001** |
| Occasionally[a] | 27 (3.5) | 25 (6.6) | 2 (0.5) | |
| Frequently | 19 (2.5) | 12 (3.2) | 7 (1.8) | |

[a] Less than two times per week or less than 6 times per month.

and lived on-campus (60.9%). Overall, 5.7% of the students were smokers, 3.5% used recreational drugs occassionally and 2.5% used recreational substance more than 3 times a week.

**Medical students.**    The mean age was 21.59 ± 1.94 of whom 50.9% were female. The majority (99.5%) were single and lived on-campus hostel accommodation (52.8%). Of them, 29 (7.7%) smoked cigarettes, 6.6% used recreational drugs more than 3 times a week and 3.2% occasionally used recreational drugs. 4% (15/767) students did not respond to questions regarding smoking.

**Nursing students.**    The mean age was 21.01 ± 1.70 of whom 89.7% female. The majority (97.7%) were single and lived on-campus hostel (68.8%). Of them, 15 (3.9%) smoked cigarettes. Only 2 (0.5%) students admitted to using recreational substance more than 3 times a week while 7 (1.8%) admitted occasional recreational substance use. A total of 11 (2.9%) students did not respond to questions regarding smoking.

## Burnout dimensions (Table 2)

The Cronbach's alpha coefficient for EX was 0.844, 0.754 for DP and 0.785 for PF.

**Medical students.**    The cut-offs calculated for medical students were **EX ≥ 20, DP ≥ 24,** and **PF ≤ 23**. The mean score for EX was 16.96 ± 6.74 for medical students. Based on this cut-off, 36.4% of medical students were categorized as having high rates of EX. The mean score for DP was 13.95 ± 7.84 for medical students with 22.2% with high rates of DP. The mean score for PF was 21.64 ± 7.21 for medical students with 53.8% with low PF.

**Nursing students.**    The cut-offs calculated for nursing students were **EX ≥ 18, DP ≥ 20,** and **PF ≤ 27**. The mean score for EX was 14.26 ± 7.30 for nursing students. Based on this cut-off, 26.8% of nursing students were categorized as having high rates of EX. The mean score for DP was 12.30 ± 5.85 for nursing students with 8.2% with high rates of DP. The mean score for PF was 25.06 ± 6.98 for nursing students with 43.3% with low PF.

## Burnout profiles (Table 2, Fig 1)

Out of 767 students, only 32.6% (n = 250) were engaged. The majority of students felt ineffective (32.5%, n = 249), and burnt-out (11.7%, n = 90). Other prevalent dimensions of burnout included disengaged and ineffective (11.0%, n = 84), overextended (8.9%, n = 68), overextended and ineffective (2.2%, n = 17) and disengaged (1.2%, n = 9).

**Table 2. Burnout dimensions and profiles among medical and nursing students.**

|  | Overall (n = 767) | Medical students (n = 379) | Nursing Students (n = 388) | p-value |
|---|---|---|---|---|
| **Burnout dimensions n (%)** |  |  |  |  |
| **High EX** | 242 (31.5) | 138 (36.4) | 104 (26.8) | 0.004 |
| **High DP** | 116 (15.1) | 84 (22.2) | 32 (8.2) | <0.001 |
| **Low PF** | 413 (53.8%) | 245 (64.6%) | 168 (43.3) | <0.001 |
| **Burnout Profiles n (%)** |  |  |  |  |
| **Engaged (↓EX, ↓DP, ↑PF)** | 250 (32.6) | 86 (22.7) | 164 (42.3) | <0.001 |
| **Disengaged (↓EX, ↑DP, ↑PF)** | 9 (1.2) | 5 (1.3) | 4 (1.0) | 0.711 |
| **Ineffective (↓EX, ↓DP, ↓PF)** | 249 (32.5) | 135 (35.6) | 114 (29.4) | 0.065 |
| **Overextended (↑EX, ↓DP, ↑PF)** | 68 (8.9) | 25 (6.6) | 43 (11.1) | 0.029 |
| **Disengaged + Ineffective (↓EX, ↑DP, ↓PF)** | 84 (11.0) | 49 (12.9) | 35 (9.0) | 0.083 |
| **Overextended + Ineffective (↑EX, ↓DP, ↓PF)** | 17 (2.2) | 15 (4.0) | 2 (0.5) | 0.001 |
| **Burnout (↑EX, ↑DP, ↑↓PF)** | 90 (11.7) | 64 (16.9) | 26 (6.7) | <0.001 |

EX: emotional exhaustion; DP: depersonalization/cynicism; PF: personal efficacy.

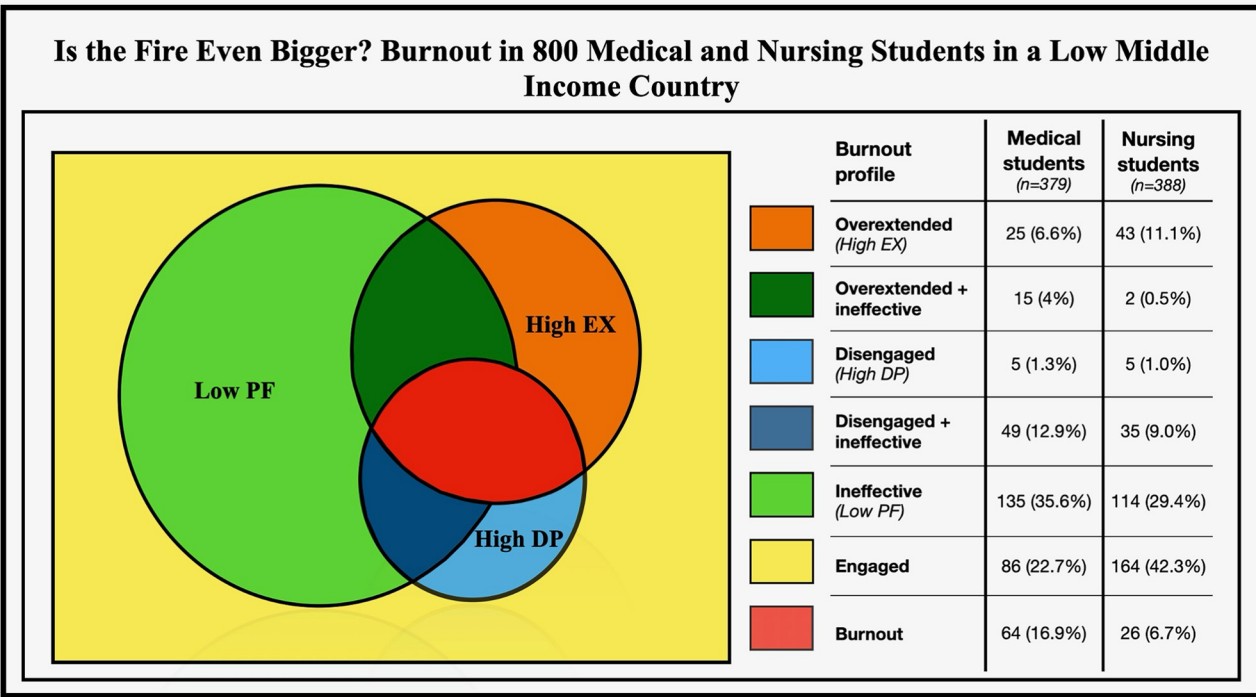

**Fig 1. Prevalence of burnout profiles based on dimensions.** EX: emotional exhaustion; DP: depersonalization; PF: personal efficacy.

**Medical students.**  Only 22.7% (n = 86) medical students felt engaged, with the majority feeling ineffective (35.6%, n = 135). A total of 16.9% (n = 64) students reported feeling burnt-out. (Table 2 & Fig 1).

**Nursing students.**  Majority of the nursing students (42.3%, n = 164) felt engaged. A third of the students reported feeling ineffective (29.4%, n = 114). A total of 26 (6.7%) students reported feeling burnout. (Table 2 & Fig 1).

## Predictors of burnout profile (Table 3, Fig 2)

Medical students were at higher risk of burnout than nursing students (OR = 2.49 [1.42, 4.38]; p<0.001) with highest risk observed in year 4 (OR = 2.47 [1.02, 5.99]; p = 0.046). Occasional drug use (OR = 1.83 [1.21, 8.49]; p = 0.017) and living in a hostel (OR = 1.64 [1.01,2.67]; p = 0.233) also appeared to be a risk factor. The probability of developing burnout increased in both medical and nursing students with increasing the year of study. However, this was more pronounced among medical students (Fig 2). Predictors of burnout for medical and nursing students independently are included in the supplement (S2 and S3 Tables).

## Discussion

Our findings show that only one-third of the student population felt engaged, as defined by the Maslach burnout inventory with low scores on the emotional exhaustion and depersonalization subscales and high scores on the personal accomplishment subscale. The remaining two-thirds experienced at least one burnout profile with the highest prevalence of the ineffective profile characterized by a reduced sense of personal efficacy. Year of training, occasional drug use and living in a hostel were risk factors associated with burnout.

**Table 3. Results of univariate and multivariable analysis of predictors of burnout among medical and nursing students.**

| Variable n (%) | Burnout | | Crude OR | p-value | Adjusted OR | p-value |
|---|---|---|---|---|---|---|
| | Yes (n = 90) | No (n = 677) | | | | |
| **Course of study** | | | | | | |
| Nursing (n = 388) | 26 (6.7) | 362 (93.3) | 1 | **<0.001** | 1 | **0.001** |
| Medicine (n = 379) | 64 (16.9) | 315 (83.1) | 2.83 (1.75, 4.57) | | 2.49 (1.42, 4.38) | |
| **Gender** | | | | | | |
| Male (n = 226) | 34 (15.2) | 192 (84.8) | 1 | **0.067** | 1 | 0.747 |
| Female (n = 541) | 56 (10.4) | 485 (89.6) | 0.65 (0.41, 1.03) | | 1.09 (0.65, 1.84) | |
| Age | 21.86 ±1.67 | 21.22 ± 1.86 | 0.82 (0.72, 0.93) | **0.002** | 0.98 (0.81, 1.17) | 0.796 |
| **Year of training** | | | | | | |
| Year 1 (n = 76) | 14 (6.6) | 197 (93.4) | 1 | **0.001** | 1 | |
| Year 2 (n = 185) | 15 (8.1) | 170 (91.9) | 1.42 (0.70, 2.86) | 0.575 | 1.20 (0.55, 2.59) | 0.650 |
| Year 3 (n = 159) | 21 (13.2) | 138 (86.8) | 1.89 (0.93, 3.85) | 0.036 | 1.80 (0.82, 3.99) | 0.145 |
| Year 4 (n = 136) | 23 (16.9) | 113 (83.1) | 3.27 (1.54, 6.95) | 0.003 | 2.47 (1.02, 5.99) | **0.046** |
| Year 5 (n = 211) | 17 (22.4) | 59 (77.6) | 4.05 (1.89, 8.71) | <0.001 | 2.17 (0.79, 5.99) | 0.133 |
| **Smoker** | | | | | | |
| No (n = 697) | 80 (11.5) | 617 (88.5) | 1 | 0.600 | | |
| Yes (n = 44) | 7 (15.9) | 37 (84.1) | 1.27 (0.52, 3.11) | | | |
| **Drug use** | | | | | | |
| No (n = 721) | 79 (11.0) | 642 (89.0) | 1 | **0.017** | 1 | |
| Occasional (n = 27) | 8 (29.6) | 19 (70.4) | 3.42 (1.45, 8.07) | | 1.83 (0.40, 8.49) | **0.017** |
| Frequent (n = 19) | 3 (15.8) | 16 (84.2) | 2.46 (0.51, 9.91) | | 2.42 (0.97, 6.07) | 0.440 |
| **Accommodation** | | | | | | |
| Home (n = 300) | 30 (10.0) | 270 (90.0) | 1 | 0.233 | 1 | **0.044** |
| Hostel (n = 467) | 60 (12.8) | 407 (87.2) | 1.33 (0.83, 2.11) | | 1.64 (1.01, 2.67) | |

Burnout among medical and nursing students is a growing concern, with prevalence rates of 35–44% for medical and 11% for nursing students [12, 24–26]. Unchecked burnout can have severe consequences, including depression, anxiety, sleep disorders, suicidal ideation, learning difficulties, thoughts of dropping out and personality changes such as lack of empathy, dishonesty, and unprofessionalism [25, 27–33]. This can ultimately impact the quality of care which these students provide in healthcare settings [34].

Several studies worldwide have explored burnout among nursing and medical students with varying results. Studies using the MBI tool typically describe burnout by high or low scores in the three dimensions (EX, DP, and PF) without detailing individual student profiles [6, 9, 10, 20, 24, 35–37]. Our findings show that proportions of students with high EX and DP are similar to previous reports, except for a notably higher proportion of students with low sense of personal efficacy.

The prevalence of the burnt-out profile (11.7%) was lower in our cohort than previously reported (37%) [38, 39]. This may be due to heterogenous methods for defining burnout, which can skew prevalence estimates [6, 9, 10, 20, 24]. Most studies using the MBI tool did not synthesize burnout profiles as instructed by the authors of the MBI tool, often using a single cut-off score (e.g., the 75[th] percentile) to categorize participants into either burnt-out or not [5, 15, 32, 40, 41]. The authors of the MBI advise against this interpretation as higher scores for questions assessing professional accomplishment go against burnout vs. questions assessing emotional exhaustion and depersonalization. Additionally, some studies used pre-determined values from literature for categorization rather than using their own population means, which

### Influence of year of study on medical vs nursing students and burnout

**Fig 2. Probability of burnout per training year for nursing (red line) and medical (black line) students.** (Dashed line represents 95% confidence interval. Adjusted for age, gender, smoking, drug use and accommodation.

is less than ideal given variation in burnout prevalence across different geographical locations [13, 15, 25, 27].

Previous studies have reported a number of factors associated with burnout. These include gender, sociodemographic variables, personality types, year of study, student loans, lack of physical activity and leisure time, mistreatment by faculty and residents, high workloads, uncertainty regarding the future, family and health issues, drug use and smoking to be associated with burnout among medical and nursing students [9, 12, 18, 19, 24, 28, 42–46]. Our study identified year of study as an independent risk factor for burnout. It should be noted that the longer you are exposed to stressors, the more likely you are to experience burnout, with extremes of burnout developing after an average exposure of around 8 years. The duration of medical (5 years) and nursing (4 years) school may not adeqautely capture the full extent of burnout that might be faced by health care professionals graduating from school.

Few studies have attempted to understand burnout among South Asian medical and nursing students where they may face a unique challenges. A study of stressors among Pakistani medical students, Shah, et al. highlighted high parental expectations, frequency of examination, worrying about the future, sleeping difficulties, and performance in periodic examinations as the common stressors leading to burnout [47, 48]. Another study showed greater burnout among medical students whose parents were doctors [48]. While many studies are inconclusive about the association of gender and burnout, some show that female students reported higher levels of stress than their male counterparts [49, 50]. Our study identified living in a hostel to be associated with a higher risk of burnout. Living away from family and not being able to enjoy their support, both emotional and for day-to day affairs may contribute. This effect may be pronounced or unique to a lower-middle-income country such as Pakistan,

where sociocultural constructs promote a close knit and more interdependent relationship among family members.

## Study implications

Findings of this study highlight the need for institutional action to prioritize student well-being. Institutions should take a multi-pronged approach in targeting student burnout. This includes readily available anonymous counseling and support services, promote a culture of safety so students can voice their concerns and respond to student feedback.

Training of faculty and residents as medical educators is crucial. The most common burn-out profile in our study was characterized by low personal efficacy, emphasizing the need for better student evaluation and feedback strategies. Policies which avoid unnecessary competition promote camaraderie among students should be encouraged.

Implementing these strategies in low-income countries like Pakistan may be challenging due to traditional practices and high competitiveness driven by limited opportunities and socioeconomic pressures. Balancing competition with motivation requires a collaborative learning environment and moving away from a grades-only evaluation system. Institutions should include diverse assessments, wellness programs focusing on mental health, stress management, and work-life balance. It is important to foster a constructive feedback culture and highlight non-academic achievements to recognize accomplishments beyond academic excellence.

## Strengths, limitations and future direction

Like any survey, this study is subject to non-response bias. This study is also a single center study which limits generalizability. However, students from all major geographical regions of Pakistan from a diverse ethnic and financial background are enrolled at the university. The use of burnout subscales allows for a better understanding of the true scope of burnout. Comparisons with data from similar centers in the country, neighboring countries, and post-COVID studies investigating burnout would further contextualize and strengthen our findings.

## Conclusion

Two-thirds of our participants suffered from at least one burnout dimension with the highest prevalence of a reduced sense of personal efficacy. Identifying the drivers of burnout unique to a lower-middle-income country is vital. Future initiatives must include cultural contextualization, longitudinal studies, and advocating for mental health policies to mitigate student burn-out. Faculty training on psychologically safe student evaluation and feedback is vital.

## Supporting information

**S1 Table. Various burnout profiles based on subscales of MBI tool.**
(DOCX)

**S2 Table. Results of univariate and multivariable analysis of predictors of burnout among medical students.**
(DOCX)

**S3 Table. Results of univariate and multivariable analysis of predictors of burnout among nursing students.**
(DOCX)

## Author Contributions

**Conceptualization:** Syed Hamza Mufarrih, Nada Qaisar Qureshi, Riaz Hussain Lakdawala, Nargis Asad, Adil Haider, Shahryar Noordin.

**Data curation:** Syed Hamza Mufarrih, Nada Qaisar Qureshi, Syeda Amrah Hashmi, Abbas Raza Syed, Muhammad Zohaib Anwar.

**Formal analysis:** Syed Hamza Mufarrih, Nada Qaisar Qureshi, Syeda Amrah Hashmi, Muhammad Zohaib Anwar.

**Investigation:** Syed Hamza Mufarrih, Nada Qaisar Qureshi, Syeda Amrah Hashmi, Abbas Raza Syed, Muhammad Zohaib Anwar, Riaz Hussain Lakdawala.

**Methodology:** Syed Hamza Mufarrih, Syeda Amrah Hashmi, Abbas Raza Syed, Muhammad Zohaib Anwar.

**Project administration:** Syed Hamza Mufarrih, Syeda Amrah Hashmi, Abbas Raza Syed, Riaz Hussain Lakdawala, Nargis Asad, Adil Haider, Shahryar Noordin.

**Supervision:** Riaz Hussain Lakdawala, Nargis Asad, Adil Haider, Shahryar Noordin.

**Writing – original draft:** Syed Hamza Mufarrih, Nada Qaisar Qureshi, Syeda Amrah Hashmi, Abbas Raza Syed.

**Writing – review & editing:** Syed Hamza Mufarrih, Nada Qaisar Qureshi, Syeda Amrah Hashmi, Abbas Raza Syed, Muhammad Zohaib Anwar.

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
