## [Decision Letter · Decision Letter 0]

23 Jan 2024

PONE-D-23-34688Is the Fire Even Bigger? Burnout in 800 Medical and Nursing Students in a Low Middle Income CountryPLOS ONE

Dear Dr. Mufarrih, 

Thank you for submitting your manuscript to PLOS ONE. After careful consideration, we feel that it has merit but does not fully meet PLOS ONE’s publication criteria as it currently stands. Therefore, we invite you to submit a revised version of the manuscript that addresses the points raised during the review process.

**The comments that were raised by the reviewers have been provided at the bottom of this letter. Take your time and revise the manuscript based on comments raised by the reviewers. We had to invite a third and fourth reviewer to assess the merit of this study for possible consideration. **

We look forward to receiving your revised manuscript.

Kind regards,

Douglas Aninng Opoku, MPH

Academic Editor

PLOS ONE

Journal Requirements:

Additional Editor Comments:

1. No sampling technique

2. Statistical analysis poorly done

3. The current burnout cut offs as used in the manuscript does not reflect the categorizations of the MBI tool used

4. How was overall burnout measured?

5. How did the authors arrive at the logistic regression since burnout measurement is a continuous variable?

6. The abstract is poorly formatted and does not meet the journal guidelines.

7. In-test citation per the journal requirement is Vancouver and not superscript as in the current format.

8. Failure to report the body that granted ethical approval for the conduct of the study.

9. Tables and figures not labelled

10. The introduction needs to be rewritten as the current state does not cover the subject fully. .

11. Improve on the grammar

12. The sample size estimation is questionable. You are dealing with two different populations but used a single population formula to estimate the sample size

13. No inclusion criteria and where the exclusion criteria are should be moved to the participants.

14. What qualifies for an incomplete survey? 70%, 80%, or less?

15. Refer to the journal's data sharing policy

16. The discussion can be improved from the current state.

Reviewers' comments:

Reviewer's Responses to Questions

**Comments to the Author**

1. Is the manuscript technically sound, and do the data support the conclusions?

Reviewer #1: No

Reviewer #2: Yes

Reviewer #3: Yes

Reviewer #4: Yes

2. Has the statistical analysis been performed appropriately and rigorously? 

Reviewer #1: No

Reviewer #2: Yes

Reviewer #3: No

Reviewer #4: Yes

3. Have the authors made all data underlying the findings in their manuscript fully available?

Reviewer #1: No

Reviewer #2: Yes

Reviewer #3: Yes

Reviewer #4: Yes

4. Is the manuscript presented in an intelligible fashion and written in standard English?

Reviewer #1: No

Reviewer #2: Yes

Reviewer #3: Yes

Reviewer #4: Yes

5. Review Comments to the Author

Reviewer #1: Clarity and Organization:

The overall structure of the discussion is poor-organized, with a not clear introduction, body, and conclusion. However, consider providing a more explicit roadmap at the beginning to guide readers through the main points.

Depth of Analysis:

The analysis of burnout among students is poor, but there's room to delve deeper into the specific factors contributing to burnout. Providing more detailed examples or case studies could enhance the depth of the discussion.

Literature Review:

While the paper touches on poor literature about student burnout, a more thorough literature review could strengthen the theoretical foundation. Consider incorporating recent studies or alternative perspectives to provide a comprehensive overview of the topic.

Methodology and Data:

If applicable, discuss the methodology used to gather information on student burnout. Providing details about the research methods or surveys employed will strengthen the credibility of the findings. analysis not done appropriate and suggest to apply new test

Causation vs. Correlation:

Be cautious about attributing causation to certain factors contributing to student burnout. Clearly differentiate between correlation and causation, and acknowledge the complexity of the issue.

Recommendations:

The discussion on potential solutions or recommendations for addressing student burnout is poorl. However, consider expanding on the feasibility and practicality of these recommendations. How easily can they be implemented, and what potential challenges might arise?

Diversity of Perspectives:

It would be beneficial to include diverse perspectives, such as different cultural or demographic influences on student burnout. This will provide a more holistic understanding of the issue.

Future Research:

Suggest avenues for future research in the field of student burnout. Identifying gaps in the current understanding and proposing areas for further investigation will contribute to the academic discourse.

Formatting and Citations:

Ensure consistent formatting and citation style throughout the paper. Check for any missing or incorrectly formatted citations to maintain scholarly integrity.

Conclusion:

The conclusion effectively poor summarizes the key points. However, consider reiterating the significance of the findings and their implications for educators, policymakers, and other stakeholders.

Reviewer #2: Dear Editor

1- References must be reviewed because the periodical in which it was published does not appear

2-The final paragraph of the inclusion criteria should be revised, it is not clear

3-the authors did not conclude the limitations of the study

Best regards

Reviewer #3: METHODS

Medical and nursing students are distinct populations in terms of sex distribution, academic ability (presumably, based on acceptance criteria in other parts of the world), academic stress exposure, duration of study etc. Seeing the sample sizes for both populations were sufficient, it may be better to analyze separately.

RESULTS

For the survey characteristics and response rate, it may be better to simple report the figures and refrain from making statements of relative size (e.g. 93.8 is greater that 91.7%) if no statistical test has been done to show that these differences are not due to chance.

Please reports results for both populations e.g. sex distribution, age etc., rather than lumping them together.

Authors assert an association between year of study and burnout. Earlier, they reported excluding first year students. The wording in table 3 (year 1 -5) is conflicting. Also it if nursing students study for fewer years, this is another reason to separate the populations because higher burnout in year 4 for a medical student may have different causes and meaning that the same for a nursing student.

Reviewer #4: The study "Is the Fire Even Bigger? Burnout in 800 Medical and Nursing Students in a Low Middle Income Country" is very interesting because it deals with a disorder that is often treated too simply, without taking into account its three-dimensional nature and the multiple variables associated with its development.

Although burnout syndrome is an occupational pathology, studies have been described in other settings (volunteers, students, etc.). It should be noted that the process by which a person reaches the most dramatic extreme of this syndrome usually involves a period of chronic exposure to stressors (considered to be an average of around 8 years), so it should be pointed out in the article that the student environment is not the most appropriate. However, the authors highlight statistically significant differences between first year students and fourth year students.

It is very interesting that different burnout profiles are described, and I would recommend explaining them “a little better” in the light of the existing literature.

Both in the initial summary and in the discussion, authors talk about the distinctions that may arise when studying the phenomenon in "a lower-middle-income country". It is suggested that this aspect be explained in more detail, especially in the discussion section.

In the Introduction section, authors talk: "previous studies have reported different prevalence rates and gender differences in burnout between medical and nursing students", it is recommended that these differences be explained in a little more detail.

It is very wise to follow the recommendations of the authors of the MBI about not using cut-off points, given the great variability by country, profession, age, etc.

The Discussion section should be more illustrative. There is no need to repeat data that have already been cited in the introduction and in the results section. Simply to interpret in the light of the results obtained in this interesting study.

Previous studies on gender, socio-demographic variables, personality types, year of study, student loans, etc. can be also discussed here.

It is pointed out as a limitation the fact that this is a study carried out in a single centre and it would be advisable to offer data from similar centres in the country, including comparisons between countries.

From the dates given, the study was conducted immediately prior to the global COVID-19 pandemic. We imagine that this data could be cross-checked with later data, given the increase in psychological pathologies resulting from the pandemic.

The fact that the authors have asked how institutions could help reduce burnout is highly commendable. The suggestions they offer are of great interest and it is recommended to develop them a "little" further, especially on "how to decrease competitiveness without abandoning the possibilities of challenge and challenge necessary for nursing and medical students".

6. PLOS authors have the option to publish the peer review history of their article (what does this mean?). If published, this will include your full peer review and any attached files.

Reviewer #1: **Yes: **Mubashir Zafar

Reviewer #2: No

Reviewer #3: **Yes: **John-Paul Omuojine

Reviewer #4: **Yes: **Santiago Gascón Santos

---

## [Author Response · Author response to Decision Letter 0]

22 Mar 2024

A point by point response to reviewers' and editor's comments has been included in the manuscript files. Briefly, the manuscript's structure has been revised to enhance clarity and coherence, incorporating a clear introduction, body, and conclusion, with a roadmap to guide readers through key points. We have expanded the analysis and broadened the literature review on the contributors to burnout, as suggested by Reviewer #1.

Further, we've added more robust details regarding our use of the Maslach Burnout Inventory, including the Cronbach coefficients relevant to the South Asian population, per Reviewer #2's concerns on methodology. The results and discussion sections have been revised to present independent findings for medical and nursing students, and we have ensured careful differentiation between correlation and causation, per Reviewer #3's advice.

As per reviewer #4's recommendations, we elaborated on burnout profiles and the unique challenges of studying this phenomenon in a lower-middle-income country. We have expanded our discussion to consider cultural and demographic influences and proposed more nuanced interventions for reducing burnout.

To the Editor's comments on methodology, we clarified that the study used a census approach, hence the traditional sample size calculation was not applicable. We have corrected the formatting of the abstract and in-text citations to comply with the journal's guidelines, and we have provided the ethical approval details and data sharing policy information.

 The manuscript has been meticulously checked to correct grammatical errors and ensure consistent citation formatting. We have addressed each point raised by the reviewers to enhance the manuscript's academic rigor and readability.

---

## [Decision Letter · Decision Letter 1]

9 Jun 2024

PONE-D-23-34688R1Is the Fire Even Bigger? Burnout in 800 Medical and Nursing Students in a Low Middle Income CountryPLOS ONE

Dear Dr. Mufarrih,

Thank you for submitting your manuscript to PLOS ONE. After careful consideration, we feel that it has merit but does not fully meet PLOS ONE’s publication criteria as it currently stands. Therefore, we invite you to submit a revised version of the manuscript that addresses the points raised during the review process.

We look forward to receiving your revised manuscript.

Kind regards,

Douglas Aninng Opoku, MPH

Academic Editor

PLOS ONE

Additional Editor Comments:

1. The authors should be specific which of the regression analysis which was employed for this study in the methods.

2. …thereby limiting our understanding of this complex syndrome. Replace ‘our’ with ‘the’

3. Depersonalization assesses a detached response toward one's studies, and Personal Accomplishment evaluates feelings of…. Check when to use capital letters

4. Individuals are classified as experiencing high levels of burnout if they score high on EE, high on DP, and either high or low on PA, per guidelines outlined by MBI. The MBI classify burnout as experiencing high EE, DP and low PA. The authors should provide a reference supporting their classification of burnout as experiencing high EE, DP and either high or low PA.

5. The authors claim they used the census approach for the recruitment of study participants. hence, sample size calculation was not applicable in their study but still go ahead to provide sample size estimation in their study. An explanation would be very much appreciated. Again, the sample size formula in the work is not appropriate especially when the authors are dealing with two populations and have provided values (p1 and p2) for the two populations.

6. There is no need to create a separate subheading for response rate. The whole sentence under the subheading can be merged with the demographic characteristics.

7. How can Institutions help to reduce burn out and promote well-being? Challenges anticipated in implementation of strategies to promote well-being: These subheadings were not assessed in the current study and could mislead readers. This could be capture under study implications.

8. The statistical analysis still needs to be clearer as the current state makes replication difficult.

9. Our findings show that only one-third of the student population was engaged….what does this means?

10. The discussion can still be improved. Especially, the authors have made some bold statements without providing references and also repeating some of the statements in the introduction in the discussion. The discussion is meant to discuss the study results by comparing with literature and giving explanations where necessary as well giving an implication of the results.

Reviewers' comments:

Reviewer's Responses to Questions

**Comments to the Author**

1. If the authors have adequately addressed your comments raised in a previous round of review and you feel that this manuscript is now acceptable for publication, you may indicate that here to bypass the “Comments to the Author” section, enter your conflict of interest statement in the “Confidential to Editor” section, and submit your "Accept" recommendation.

Reviewer #5: (No Response)

Reviewer #6: (No Response)

2. Is the manuscript technically sound, and do the data support the conclusions?

Reviewer #5: Yes

Reviewer #6: Yes

3. Has the statistical analysis been performed appropriately and rigorously? 

Reviewer #5: Yes

Reviewer #6: Yes

4. Have the authors made all data underlying the findings in their manuscript fully available?

Reviewer #5: Yes

Reviewer #6: Yes

5. Is the manuscript presented in an intelligible fashion and written in standard English?

Reviewer #5: Yes

Reviewer #6: Yes

6. Review Comments to the Author

Reviewer #5: Many thanks to adding to the literature.

It is now obvious that healthcare workers, regardless of geography, are affected by burnout. It's time to move on finding solutions.

The authors have done a great job of rewriting their paper. All of my comments are to improve readability and flow.

27: Change to ...67.5% (517/727) of medical and nursing students... It changes the emphasis of the sentence onto the results.

39: Change ...We aim to describe... to ... We describe... You have described after all.

48: I'm conflicted about the use of the shorthand to describe the parts of burnout. Healthcare workers reading the paper will read PE as Pulmonary Embolism so using it for personal efficacy increases the cognitive load of the reader.

56: Change ...stidents were engaged... to ...students felt engaged...

61: Change ...other risk factors... to ...other risk factors for burnout...

63: Change ...Two-thirds of our participants suffered from at least one burnoiut dimension... to ...Two-thirds of our participants experienced at least one dimension of burnout...

71: Change ...the three dimensions of burnout... to ...the three burnout dimensions...

74: Change ...manifesting in a myriad... to ...with many...

75: What exactly do you mean by metabolic disorders? Diabetes? Cardiovascular disease?

83: Reword ...The pressures of competition for higher performance... as it doesn't make sense as a sentence. Perhaps just say ...The pressures of competition...

85: Change ...the vulnerability of students... to ...a student's vulnerability...

94: Remove the word ...Furthermore... - it adds nothing to the sentence

95: Remove ...often...

96. Split into two sentences ....single-measures. These oversimplify...

97. Changing ...Using... to ...Using the results of the MBI...

101. Change ...utilized... for used. Less syllables increases readability

107. Change ... The aim of our study... to ...Our study aimed...

115. Remove ...a period of... It's reductive

122. Replace ...Students... with ...Those... You've already used the word students in the previous sentence

128. Change ...The e-mail informed... with ...Students were informed...

130. Change ...had their e-mails removed... to ...students were removed...

131. ...The survey was incentivized... should be a new paragraph.

145. Change ...MBI burnout tool... to ... The MBI tool. Remember what MBI stands for. If you write out your sentence it would read Maslach Burnout Inventory Burnout Tool.

148. Emotional Exhaustion describes feelings of ... it doesn't measure them.

152-154. This could be clearer.

160. Incorrect spelling of Categorized

179-181. Would sound better as ...and ensure findings accurately respresent the populations attribute prevalence, we calculated the minimal sample size using the formula...

200. Easier to say .. in each group... rather than ... in both groups together and seperately...

231. Change ...the data cannot be made available publically... to ...is not available publically...

241. Make a new sentence of ...The average completion time... Though I would suggest removing this as it is irrelevent data that adds nothing to your narrative. You don't suggest it is a statistically significant difference after all.

253. Should be ...The majority were single and 60.9% lived on campus...

254. What does endorsed mean? Does it mean that said drugs were okay or does it mean they used them? Clearer wording would be ... 3.5% used recreational substances occassioanlly and 2.5% used it more than X times a week... You need to let the reader know what the difference. isbetween occassional and frequent use.

259. It's clearer if written as ...4% (15/767) students did not dicslose their smoking status...

263. Again, you use the term endorse when it's meaning is not clear.

271. Change to ...Based on this cut-off...

288-289. It is more powerful to say ...The majority of medical students felt ineffective and burnt out...

292. Again, the medical students felt engaged rather than were engaged.

295. Same again.

304. Change ...compared to... to ...than...

306. Would be clearer to start a new sentence as ... Occasional drug use also appeared to be a risk factor...

313. Students ...felt... engaged

318. Change ...risk factors found to be associated... to ... were associated with...

336. Change the sentence order to make it more powerful. ...The prevalence of the burnt out profile was lower in our cohort than previously reported...

346. ...the authors of the MBI advise against this as...

354. You can make this clearer by saying ... Previous studies have reported a number of factors associated with burnout. These include...

359. This could be more cleqrly expressed as ...The longer youy are exposed to stressors, the more likely you are to experience burnout...

363. Use less words ..healthcare professionals readuating...

369. Spelling of interdependent

377. Did you capture any data around parental occupation in your cohort?

379. ...Many... not ...Multiple...

380. ...While many are inconclusive...

383. ...Use... not ...Utilization...

389. You can make this punchier. Try ...Institutions should take a multi-promged approach in targetting student burnout. This includes readily available, anonymous counselling and support services...

394. Don't hide a new argument in the middle of the paragraph. New point = new para

405. Think of the international reader. What is unique about your circumstances in Pakistan? What do you mean by 'significant youth bulge'?

423. You want to close strongly. ...Identifying the drivers of burnout unique to LMICs is vital. Future initiatives must include...

426-427. This is a very soft ending given you state earlier that it is key. Perhaps ...Faculty training on psychologically safe student evaluation and feedback is vital...

Again, I want to reiterate that I am reviewing this as a native English speaker and my suggestions are only to help you have a better paper.

Dr Andrew Tagg

Reviewer #6: Dear Authors,

Thank you for your interest in this important research area. This is very relevant as has potential impact on the quality of care which these students will ultimately deliver in the healthcare setup. Your report is clearly presented, however, I have minor observations:

Consider rephrasing lines 70-72; “Therefore, it is imperative to identify and address the ongoing stressors associated…..”

Line 107: If I may ask, why was the survey incentive awarded to only the 5th participant and not all survey participants?

7. PLOS authors have the option to publish the peer review history of their article (what does this mean?). If published, this will include your full peer review and any attached files.

Reviewer #5: **Yes: **Dr Andrew Tagg

Reviewer #6: No

---

## [Author Response · Author response to Decision Letter 1]

12 Jun 2024

Editor Comments: 

1. Thank you for your feedback. In response, we have clarified the specific regression analyses used in our study. We employed both bivariate logistic regression to explore individual relationships between demographic variables and burnout, and multivariable logistic regression to identify independent predictors of burnout, using SPSS Statistics (version 29.0). These details have been added to the methods section for clarity.

2, 3. Thank you for your comment. These have been corrected. 

4. Thank you for your comment. As you aptly identified, this is per guidelines by the MBI, the authors of the survey. We have referenced the survey in the text following this statement. 

5. Thank you for your insightful comments. We have revised the manuscript to exclude the sample size calculation section, as a census approach was utilized, making sample size calculation unnecessary. This should clarify our recruitment strategy.

6. Thank you for your comment. This has been changed.

7. Thank you for your comment. The heading for challenges was originally added as per a reviewer’s suggestion in the first review. To best consolidate these two comments, the headings have been removed but the content has been retained. 

8. Thank you for your feedback. We have revised the statistical analysis section to provide more detailed and clear descriptions of the methods used, to improve transparency and ease of replication. The revised section now includes specific tests used, the rationale for their selection, and detailed steps of the analysis process.

9. Thank you for your comment. Results show that only 32.6% of the students felt engaged (Table 2) which is defined by the Maslach burnout inventory as low scores on the emotional exhaustion and depersonalization subscales and high scores on the personal accomplishment subscales. We have added this explanation to the text as well. 

 10. Thank you for your feedback. The discussion appeared a bit scattered. We have revised and simplified this section while doing our best to retain the additions that were requested by reviewers in the previous review cycle. We have organized it as follows: 

-Summary of findings 

-Significance and consequences of burnout 

-Comparison of subscales scores to previous literature 

-Comparison of burnout profiles to previous literature with explanation of differences 

-Comparison of risk factors associated with burnout in general and specific to the South Asian population to previous literature 

-Study implications

-Strengths/limitations and future direction

Reviewer#5: 

Thank you for your constructive comments. 

1,2. These have been changed. 

3. We agree, it is confusing, and PE does cognitively default to pulmonary embolism. For this, we initially contemplated referring to personal efficacy as personal accomplishment and abbreviating as PA. 

However, the authors of the MBI tool have described this specifically as personal efficacy and explain that this refers to a student’s satisfaction with their accomplishments rather than accomplishments alone. https://www.mindgarden.com/313-mbi-general-survey-for-students

We have now changed PE to PF to fix this. 

4-8. These have been changed. 

9. Thank you for your comment. Metabolic disorders are referring to diabetes and metabolic syndrome. This has been added to the text. 

10-26. These have been changed. 

27. Thank you for your comment. The sample size section was deleted per editors comment therefore this change couldn’t be made.

28-54. These have been changed. 

55.Thank you for your question. The authors were referring to a greater proportion of younger people in Pakistan, given lower life expectancy and higher birth rates compared with a developed country. However, this paragraph has now been revised to simply and organize the discussion as suggested by the editor comments and this phrase is now excluded. 

56,57. These have been changed. 

Reviewer#6

1. Thank you for your comment. This sentence has been revised. 

2. Thank you for your question. This was mainly due to limited funds for the study and the budget did not allow for every participant to receive the study incentive.

---

## [Decision Letter · Decision Letter 2]

3 Jul 2024

Is the Fire Even Bigger? Burnout in 800 Medical and Nursing Students in a Low Middle Income Country

PONE-D-23-34688R2

Dear Dr. Mufarrih,

We’re pleased to inform you that your manuscript has been judged scientifically suitable for publication and will be formally accepted for publication once it meets all outstanding technical requirements.

Kind regards,

Douglas Aninng Opoku, MPH

Academic Editor

PLOS ONE

Additional Editor Comments (optional):

I want to congratulate the authors for taking their time to address all comments raised by the reviewers in the previous review. I have recommended that your manuscript be accepted for publication. The authors captured exclusion criteria under subheading 'eligibility criteria'. Move that statement and create another subheading for exclusion criteria and place that statement there.

Reviewers' comments:

Reviewer's Responses to Questions

**Comments to the Author**

1. If the authors have adequately addressed your comments raised in a previous round of review and you feel that this manuscript is now acceptable for publication, you may indicate that here to bypass the “Comments to the Author” section, enter your conflict of interest statement in the “Confidential to Editor” section, and submit your "Accept" recommendation.

Reviewer #5: All comments have been addressed

Reviewer #6: All comments have been addressed

2. Is the manuscript technically sound, and do the data support the conclusions?

Reviewer #5: (No Response)

Reviewer #6: (No Response)

3. Has the statistical analysis been performed appropriately and rigorously? 

Reviewer #5: (No Response)

Reviewer #6: N/A

4. Have the authors made all data underlying the findings in their manuscript fully available?

Reviewer #5: (No Response)

Reviewer #6: (No Response)

5. Is the manuscript presented in an intelligible fashion and written in standard English?

Reviewer #5: (No Response)

Reviewer #6: (No Response)

6. Review Comments to the Author

Reviewer #5: (No Response)

Reviewer #6: (No Response)

7. PLOS authors have the option to publish the peer review history of their article (what does this mean?). If published, this will include your full peer review and any attached files.

Reviewer #5: **Yes: **Dr Andrew James Tagg

Reviewer #6: **Yes: **Okyere Daniel

---

## [Editor Report · Acceptance letter]

17 Jul 2024

PONE-D-23-34688R2 

PLOS ONE

Dear Dr. Mufarrih, 

I'm pleased to inform you that your manuscript has been deemed suitable for publication in PLOS ONE. Congratulations! Your manuscript is now being handed over to our production team.

Kind regards, 

on behalf of

Dr. Douglas Aninng Opoku 

Academic Editor

PLOS ONE